# Offline Behavior Distillation

**Shiye Lei**
School of Computer Science
The University of Sydney
shiye.lei@sydney.edu.au

**Sen Zhang**
School of Computer Science
The University of Sydney
sen.zhang@sydney.edu.au

**Dacheng Tao**
College of Computing & Data Science
Nanyang Technological University
dacheng.tao@ntu.edu.sg

## Abstract

Massive reinforcement learning (RL) data are typically collected to train policies offline without the need for interactions, but the large data volume can cause training inefficiencies. To tackle this issue, we formulate offline behavior distillation (OBD), which synthesizes limited expert behavioral data from sub-optimal RL data, enabling rapid policy learning. We propose two naive OBD objectives, DBC and PBC, which measure distillation performance via the decision difference between policies trained on distilled data and either offline data or a near-expert policy. Due to intractable bi-level optimization, the OBD objective is difficult to minimize to small values, which deteriorates PBC by its distillation performance guarantee with quadratic discount complexity $\mathcal{O}(1/(1-\gamma)^2)$. We theoretically establish the equivalence between the policy performance and action-value weighted decision difference, and introduce action-value weighted PBC (Av-PBC) as a more effective OBD objective. By optimizing the weighted decision difference, Av-PBC achieves a superior distillation guarantee with linear discount complexity $\mathcal{O}(1/(1-\gamma))$. Extensive experiments on multiple D4RL datasets reveal that Av-PBC offers significant improvements in OBD performance, fast distillation convergence speed, and robust cross-architecture/optimizer generalization. The code is available at https://github.com/LeavesLei/OBD.

## 1 Introduction

Due to the costs and dangers associated with interactions in reinforcement learning (RL), learning policies from pre-collected RL data has become increasingly popular [Levine et al., 2020]. Consequently, numerous offline RL datasets have been constructed [Fu et al., 2020]. However, these offline data are typically massive and collected by sub-optimal or even random policies, leading to inefficiencies in policy training. Inspired by dataset distillation (DD) [Wang et al., 2018, Zhao et al., 2021, Lei and Tao, 2024], which synthesizes a small number of training images while preserving model training effects, we further investigate the following question: *Can we distill vast sub-optimal RL data into limited expert behavioral data?* Achieving this would enable rapid offline policy learning via behavioral cloning (BC) [Pomerleau, 1991], which can (1) reduce the training cost and enable green AI; (2) facilitate downstream tasks by using distilled data as prior knowledge (*e.g.* continual RL [Gai et al., 2023], multi-task RL [Yu et al., 2021], efficient policy pretraining [Goecks et al., 2019], offline-to-online fine-tuning [Zhao et al., 2022]); and (3) protect data privacy [Qiao and Wang, 2023].

Unlike DD whose objective is prediction accuracy and directly obtainable from real data, the policy performance in RL is measured by the expected return through interactions with environment. In an

38th Conference on Neural Information Processing Systems (NeurIPS 2024).

offline paradigm, where direct interaction with environment is not possible, a metric based on RL data is necessary to guide the RL data distillation. Therefore, we formalize the offline behavior distillation (OBD): a limited set of behavioral data, comprising (`state`, `action`) pairs, is synthesized from sub-optimal RL data, so that policies trained on the compact synthetic dataset by BC can achieve small OBD objective loss, which incarnates high return when deploying policies in the environment.

The key obstacle for OBD is constructing a proper objective that efficiently and accurately estimates the policy performance based on the sub-optimal offline dataset, allowing for a rational evaluation of the distilled data. To this end, data-based BC (DBC) and policy-based BC (PBC) present two naive OBD objectives. Specifically, DBC reflects the policy performance by measuring the mismatch between the policy decision and vanilla offline data. Leveraging existing offline RL algorithms that can extract near-optimal policies from sub-optimal data [Levine et al., 2020], PBC improves upon DBC by correcting actions in offline data using a near-optimal policy before measuring the decision difference. However, due to the complex bi-level optimization in OBD, the objectives are difficult to minimize effectively, resulting in an inferior distillation performance guarantee with the *quadratic* discount complexity $\mathcal{O}(1/(1-\gamma)^2)$ for PBC (Theorem 1). We tackle this problem and propose the action-value weighted PBC (Av-PBC) as the OBD objective with superior distillation guarantee by taking inspirations from our theoretical findings. Concretely, we theoretically prove the equivalence between the policy performance gap and the action-value weighted decision difference (Theorem 2). Then, by optimizing the weighted decision difference, we can obtain a much tighter distillation performance guarantee with *linear* discount complexity $\mathcal{O}(1/(1-\gamma))$ (Corollary 1). Consequently, we weigh PBC with the simple action value, introducing Av-PBC as the OBD objective.

Extensive experiments on nine datasets of D4RL benchmark [Fu et al., 2020] with multiple environments and data qualities illustrate that our Av-PBC remarkably promotes the OBD performance, which is measured by normalized return, by $82.8\%$ and $25.7\%$ compared to baselines of DBC and PBC, respectively. Moreover, Av-PBC has a significant convergence speed and requires only a quarter of distillation steps compared to DBC and PBC. By evaluating the synthetic data in terms of different network architectures and training optimizers, we show that distilled datasets possess decent cross-architecture/optimizer performance. Apart from evaluations on single policy, we also investigate policy ensemble performance by training multiple policies on the synthetic dataset and combining them to generate actions. The empirical findings demonstrate that the ensemble operation can significantly enhance the performance of policies trained on Av-PBC-distilled data by $25.8\%$.

Our contributions can be summarized as:

- We formulate the offline behavior distillation problem, and present two naive OBD objectives of DBC and the improved PBC;
- We demonstrate the unpleasant distillation performance guarantee of $\mathcal{O}(1/(1-\gamma)^2)$ for PBC, and theoretically derive a novel objective of Av-PBC that has much tighter performance guarantee of $\mathcal{O}(1/(1-\gamma))$;
- Extensive experiments on multiple offline RL datasets verify significant improvements on OBD performance and speed by Av-PBC.

## 2 Related works

**Offline RL** Data collection can be both hazardous (*e.g.* autonomous driving) and costly (*e.g.* healthcare) with the online learning paradigm of RL. To alleviate the online interaction, offline RL has been developed to learn the policy from a pre-collected dataset gathered by sub-optimal behavior policies [Lange et al., 2012, Fu et al., 2020]. However, the offline paradigm limits exploration and results in the distributional shift problem: (1) the state distribution discrepancy between learned policy and behavior policy at test time; and (2) only in-dataset state transitions are sampled when conducting Bellman backup [Bellman, 1966] during the training period [Levine et al., 2020]. Various offline RL algorithms have been proposed to mitigate the distributional shift problem. Fujimoto and Gu [2021], Tarasov et al. [2024] introduce policy constrain that control the discrepancy between learned policy and behavior policy. To address the problem of over-optimistic estimation on out-of-distribution actions, Kumar et al. [2020], Nakamoto et al. [2023], Kostrikov et al. [2022] propose to regularize the learned value function for conservative Q learning. Moreover, ensemble approaches have also proven effective in offline RL [An et al., 2021]. Readers can refer to [Tarasov et al., 2022] for a detailed comparison of offline RL methods. Albeit these advancements, the offline dataset is extremely large

(million-level) and contains sensitive information (*e.g.* medical history) [Qiao and Wang, 2023], necessitating consideration of training efficiency, data storage, and privacy concerns. To address these issues, we distill a small behavioural dataset from vast subpar offline RL data to enable efficient policy learning via BC.

**Dataset Distillation** Given the resource constraints in era of big data, numerous approaches have focused on improving learning efficiency through memory-efficient model [Han et al., 2016, Jing et al., 2021] and effective data utilization [Mirzasoleiman et al., 2020, Jing et al., 2023, Lei et al., 2023]. Recently, dataset distillation (DD) has emerged as a promising technique for condensing large real datasets into significantly smaller synthetic ones, such that models trained on these tiny synthetic datasets achieve comparable generalization performance to those trained on large original datasets [Sachdeva and McAuley, 2023, Yu et al., 2024, Lei and Tao, 2024]. This approach addresses key issues such as training inefficiency, data storage limitations, and data privacy concerns. There are two primary frameworks for DD: the meta-learning framework, which formulates dataset distillation as a bi-level optimization problem [Wang et al., 2018, Deng and Russakovsky, 2022], and the matching framework, which matches the synthetic and real datasets in terms of gradient [Zhao et al., 2021, Zhao and Bilen, 2021], feature [Zhao and Bilen, 2023, Wang et al., 2022], or training trajectory [Cazenavette et al., 2022, Cui et al., 2023].

While most DD methods focus on image data, Lupu et al. [2024] propose behavior distillation (BD), extending DD to online RL regime. In (online) BD, a small number of state-action pairs are synthesized for fast BC training by (1) directly computing policy returns through online interactions; and (2) estimating the meta-gradient *w.r.t.* synthetic data via evolution strategies (ES) [Salimans et al., 2017]. We underline that our OBD is not an extension of online BD, but rather a novel and parallel field because of different objectives that incur distinct challenges: (1) online BD uses the ground truth objective, *i.e.*, policy return, by sampling many long episodes from environments. As a result, backpropagating the meta-gradient of return *w.r.t.* synthetic data is extremely inefficient, and Lupu et al. [2024] tackle the challenge by estimating meta-gradient with the zero-order algorithm of ES; and (2) OBD objective solely relies on offline data instead of long episode sampling, thereby making meta-gradient backpropagation relatively efficient and feasible, and the primary obstacle for OBD lies in designing an appropriate objective that accurately reflects the policy performance.

## 3 Preliminaries

**Reinforcement Learning** The problem of reinforcement learning can be described as the Markov decision process (MDP) $\langle \mathcal{S}, \mathcal{A}, \mathcal{T}, r, \gamma, d^0 \rangle$, where $\mathcal{S}$ is a set of states $s \in \mathcal{S}$, $\mathcal{A}$ is the set of actions $a \in \mathcal{A}$, $\mathcal{T}(s'|s,a)$ denotes the transition probability function, $r(s,a)$ is the reward function, $\gamma \in (0,1)$ is the discount factor, and $d^0(s)$ is the initial state distribution [Sutton and Barto, 2018]. We assume that the reward function is bounded by $R_{\max}$, *i.e.*, $r(s,a) \in [0, R_{\max}]$ for all $(s,a) \in \mathcal{S} \times \mathcal{A}$. The objective of RL is to learn a policy $\pi(a|s)$ that maximizes the long-term expected return $J(\pi) = \mathbb{E}_\pi \left[ \sum_{t=0}^\infty \gamma^t r_t \right]$, where $r_t = r(s_t, a_t)$ is the reward at $t$-step, and $\gamma$ usually is close to 1 to consider long-horizon rewards in the most RL tasks. We define $d_\pi^t(s) = \Pr(s_t = s; \pi)$ and $\rho_\pi^t(s,a) = \Pr(s_t = s, a_t = a; \pi)$ as $t$-th step state distribution and state-action distribution, respectively. Then, the discounted stationary state distribution $d_\pi(s) = (1-\gamma) \sum_{t=0}^\infty \gamma^t d_\pi^t(s)$, and the discounted stationary state-action distribution $\rho_\pi(s,a) = (1-\gamma) \sum_{t=0}^\infty \gamma^t \rho_\pi^t(s,a)$. Intuitively, the state (state-action) distribution depicts the overall "frequency" of visiting a state (state-action) with $\pi$. The action-value function of $\pi$ is $q_\pi(s,a) = \mathbb{E}_\pi \left[ \sum_{t=0}^\infty \gamma^t r_t \mid s_0 = s, a_0 = a \right]$, which is the expected return starting from $s$, taking the action $a$. Since $r_t \geq 0$, we have $q_\pi(s,a) \geq 0$ for all $(s,a)$.

Instead of interacting with the environment, offline RL learns the policy from a sub-optimal offline dataset $\mathcal{D}_{\text{off}} = \{(s_i, a_i, s_i', r_i)\}_{i=1}^{N_{\text{off}}}$ with specially designed Bellman backup [Levine et al., 2020]. Although $\mathcal{D}_{\text{off}}$ is normally collected by sub-optimal behavior policies, offline RL algorithms can recapitulate a near-optimal policy $\pi^*$ and value function $q_{\pi^*}$ from $\mathcal{D}_{\text{off}}$.

**Behavioral Cloning** [Pomerleau, 1991] can be regarded as a special offline RL algorithm and only copes with high-quality data. Given the expert demonstrations $\mathcal{D}_{\text{BC}} = \{(s_i, a_i)\}_{i=1}^{N_{\text{BC}}}$, the policy network $\pi_\theta$ parameterized by $\theta$ is trained by cloning the behavior of the expert dataset $\mathcal{D}_{\text{BC}}$ in a supervised manner: $\min_\theta \ell_{\text{BC}}(\theta, \mathcal{D}_{\text{BC}}) := \mathbb{E}_{(s,a) \sim \mathcal{D}_{\text{BC}}} \left[ (\pi_\theta(a|s) - \hat{\pi}^*(a|s))^2 \right]$, where $\hat{\pi}^*(a|s) =$

$\frac{\sum_{i=1}^{N_{\text{BC}}} \mathbb{I}(s_i=s, a_i=a)}{\sum_{i=1}^{N_{\text{BC}}} \mathbb{I}(s_i=s)}$ is an empirical estimation based on $\mathcal{D}_{\text{BC}}$. Compared to general offline RL algorithms that deal with subpar 4-tuples of $\mathcal{D}_{\text{off}}$, BC only handles expert 2-tuples, while it has better convergence speed due to the supervised paradigm. This paper aims to distill massive sub-optimal 4-tuples into a few expert 2-tuples, thereby enabling rapid policy learning via BC.

## 3.1 Problem Setup

We first introduce behavior distillation [Lupu et al., 2024] that aims to synthesize few data points $\mathcal{D} = \mathcal{D}_{\text{syn}} = \{(s_i, a_i)\}_{i=1}^{N_{\text{syn}}}$ with small $N_{\text{syn}}$ from the environment, so the policy trained on $\mathcal{D}_{\text{syn}}$ has a large expected return $J$. The problem of behavior distillation can be formalized as follows:

$$\mathcal{D}_{\text{syn}}^* = \arg\max_{\mathcal{D}} J\left(\pi_{\theta(\mathcal{D})}\right) \quad \text{s.t.} \quad \theta(\mathcal{D}) = \arg\min_{\theta} \ell_{\text{BC}}(\theta, \mathcal{D}). \tag{1}$$

During behavior distillation, the return $J$ is directly estimated by the interaction between policy and environment. However, in the offline setting, the environment can not be touched, and only the previously collected dataset $\mathcal{D}_{\text{off}}$ is provided. Hence, we employ $\mathcal{H}(\pi_\theta, \mathcal{D}_{\text{off}})$ as a surrogate loss to estimate the policy performance of $\pi_\theta$ given the offline data $\mathcal{D}_{\text{off}}$ without interactions with the environment. Then, by setting $N_{\text{syn}} \ll N_{\text{off}}$, *offline behavior distillation* can be formulated as below:

$$\mathcal{D}_{\text{syn}}^* = \arg\min_{\mathcal{D}} \mathcal{H}\left(\pi_{\theta(\mathcal{D})}, \mathcal{D}_{\text{off}}\right) \quad \text{s.t.} \quad \theta(\mathcal{D}) = \arg\min_{\theta} \ell_{\text{BC}}(\theta, \mathcal{D}). \tag{2}$$

## 3.2 Backpropagation through Time

The formulation of offline behavior distillation is a bi-level optimization problem: the inner loop optimizes the policy network parameters based on the synthetic dataset with BC by multiple iterations of $\{\theta_1, \theta_2, \cdots, \theta_T\}$. During the outer loop iteration, synthetic data are updated by minimizing the surrogate loss $\mathcal{H}$. With the nested loop, the synthetic dataset gradually converges to one of the optima. This bi-level optimization can be solved by backpropagation through time (BPTT) [Werbos, 1990]:

$$\nabla_{\mathcal{D}} \mathcal{H} = \frac{\partial \mathcal{H}}{\partial \mathcal{D}} = \frac{\partial \mathcal{H}}{\partial \theta^{(T)}} \left( \sum_{k=0}^{k=T} \frac{\partial \theta^{(T)}}{\partial \theta^{(k)}} \cdot \frac{\partial \theta^{(k)}}{\partial \mathcal{D}} \right), \quad \text{and} \quad \frac{\partial \theta^{(T)}}{\partial \theta^{(k)}} = \prod_{i=k+1}^{T} \frac{\partial \theta^{(i)}}{\partial \theta^{(i-1)}}. \tag{3}$$

Although BPTT provides a feasible solution to compute the meta-gradient for OBD, the objective $H$ is hardly minimized to near zero in practice owing to the severe complexity and non-convexity of bi-level optimization [Wiesemann et al., 2013].

# 4 Methods

The key challenge in OBD is *determining an appropriate objective loss $\mathcal{H}(\pi_\theta, \mathcal{D}_{off})$ to estimate the performance of $\pi_\theta$*. While policy performance could be naturally estimated using episode return by learning a MDP environment from $\mathcal{D}_{\text{off}}$, as done in model-based offline RL [Kidambi et al., 2020], this approach is computationally expensive. Apart from the considerable time required to sample the episode for evaluation, the corresponding gradient computation is also inefficient: although Policy Gradient Theorem $\frac{\partial J}{\partial \theta} = \sum_s d_\pi(s) \sum_a q_\pi(s, a) \nabla_\theta \pi_\theta(a|s)$ provides a way to compute meta-gradients [Sutton and Barto, 2018], the gradient estimation often exhibits high variance due to the lack of information *w.r.t.* $d_\pi(s)$ and $q_\pi(s, a)$.

## 4.1 Data-based and Policy-based BC

Compared to both sampling and gradient computation inefficiency of policy return, directly using $\mathcal{D}_{\text{off}}$ is a more feasible way to estimate the policy performance in OBD, and a natural option is BC loss, *i.e.*, $\mathcal{H}(\pi_\theta, \mathcal{D}_{\text{off}}) = \ell_{\text{BC}}(\theta, \mathcal{D}_{\text{off}})$, which we refer to as **data-based BC (DBC)**. However, as $\mathcal{D}_{\text{off}}$ is collected by sub-optimal policies, DBC hardly evaluates the policy performance accurately.

Benefiting from offline RL algorithms, we can extract the near-optimal policy $\pi^*$ and corresponding value function $q_{\pi^*}$ from $\mathcal{D}_{\text{off}}$ via carefully designed Bellman updates. Consequently, a more rational choice is to correct actions in $\mathcal{D}_{\text{off}}$ with $\pi^*$, leading to $\mathcal{H}(\pi, \mathcal{D}_{\text{off}}) = \mathbb{E}_{s \sim \mathcal{D}_{\text{off}}} [D_{\text{TV}}(\pi^*(\cdot|s), \pi(\cdot|s))]$, where $D_{\text{TV}}(\pi^*(\cdot \mid s), \pi(\cdot \mid s)) = \frac{1}{2} \sum_{a \in \mathcal{A}} [|\pi^*(a|s) - \pi(a|s)|]$ is the total variation (TV) distance

that measures the decision difference between $\pi^*$ and $\pi$ at state $s$, and we term this metric as **policy-based BC (PBC)**. With the exemplar $\pi^*$, offline behavior distillation performance $J(\pi)$, where $\pi$ is trained on $\mathcal{D}_{\text{syn}}$, can be guaranteed by the following theorem.

**Theorem 1** (Theorem 1 in [Xu et al., 2020]). *Given two policies of $\pi^*$ and $\pi$ with $\mathbb{E}_{s\sim d_{\pi^*}(s)}\left[D_{\text{TV}}\left(\pi^*(\cdot|s),\pi(\cdot|s)\right)\right] \le \epsilon$, we have $|J(\pi^*) - J(\pi)| \le \frac{2R_{\max}}{(1-\gamma)^2}\epsilon$.*

**Remark 1.** *The proof of Theorem 1 does not necessitate that $\pi^*$ is superior to $\pi$, and thus substituting $s \sim d_{\pi^*}(s)$ in $\mathbb{E}_{s\sim d_{\pi^*}(s)}\left[D_{\text{TV}}\left(\pi^*(\cdot|s),\pi(\cdot|s)\right)\right] \le \epsilon$ with $s \sim d_\pi(s)$ does not alter the outcome.*

Theorem 1 elucidates that $\pi$ has close performance to the good policy $\pi^*$ as long as they act similarly, and $J(\pi) \to J(\pi^*)$ if their decision difference $D_{\text{TV}}\left(\pi^*(\cdot\mid s),\pi(\cdot\mid s)\right) \to 0$. This is optimistic for the conventional BC setting where the loss can be easily optimized to near zero. However, because of intractable bi-level optimization, the empirical objective $\epsilon$ is rarely decreased to small values in OBD. According to [Xu et al., 2020], the upper bound in Theorem 1 is tight as quadratic discount complexity $\mathcal{O}(1/\left(1-\gamma\right)^2)$ is inevitable in the worst-case, implying that the distillation performance guarantee collapses quickly as the PBC objective increases. To this end, a more effective OBD objective should be considered to ensure stronger distillation guarantees.

## 4.2 Action-value weighted PBC

The preceding analysis highlights the inferior distillation guarantee of $\mathcal{O}(1/(1-\gamma)^2)$ with PBC. To establish a superior OBD objective, we prove the equivalence between the performance gap of $J(\pi^*) - J(\pi)$ and action-value weighted $\pi^*(a|s) - \pi(a|s)$ (Theorem 2). By optimizing the weighted decision difference, the performance gap can be non-vacuously bounded with a reduced discount complexity of $\mathcal{O}(1/(1-\gamma))$ (Corollary 1). Motivated by these theoretical insights, we propose action-value weighted PBC as the OBD objective for a tighter distillation performance guarantee.

**Theorem 2.** *For any two policies $\pi$ and $\pi^*$, we have*

$$J(\pi^*) - J(\pi) = \frac{1}{1-\gamma}\mathbb{E}_{s\sim d_\pi(s)}\left[q_{\pi^*}(s,\cdot)\left(\pi^*(\cdot|s)-\pi(\cdot|s)\right)\right], \tag{4}$$

*where the dot notation $(\cdot)$ is a summation over the action space, i.e., $q_{\pi^*}(s,\cdot)\left(\pi^*(\cdot|s)-\pi(\cdot|s)\right) = \sum_{a\in\mathcal{A}} q_{\pi^*}(s,a)\left(\pi^*(a|s)-\pi(a|s)\right)$.*

**Proof Sketch.** (1) With RL definitions, we represent $J(\pi^*) - J(\pi)$ by

$$J(\pi^*) - J(\pi) = \mathbb{E}_{s\sim d_{\pi^*}^0(s)}\left[q_{\pi^*}(s,\cdot)\left(\pi^*(\cdot|s)-\pi(\cdot|s)\right)\right] + \mathbb{E}_{\rho_\pi^1(s,a)}\left[q_{\pi^*}(s,a)-q_\pi(s,a)\right];$$

(2) then we prove the iterative formula *w.r.t.* $\mathbb{E}_{\rho_\pi^n(s,a)}\left[q_{\pi^*}(s,a)-q_\pi(s,a)\right]$:

$$\mathbb{E}_{\rho_\pi^n(s,a)}\left[q_{\pi^*}(s,a)-q_\pi(s,a)\right]$$
$$=\gamma\mathbb{E}_{s\sim d_\pi^{n+1}(s)}\left[q_{\pi^*}(s,\cdot)\left(\pi^*(\cdot|s)-\pi(\cdot|s)\right)\right] + \gamma\mathbb{E}_{\rho_\pi^{n+1}(s,a)}\left[q_{\pi^*}(s,a)-q_\pi(s,a)\right];$$

(3) integrating the two equations above yields the desired result

$$J(\pi^*) - J(\pi) = \sum_{t=0}^\infty \gamma^t \mathbb{E}_{s\sim d_\pi^t(s)}\left[q_{\pi^*}(s,\cdot)\left(\pi^*(\cdot|s)-\pi(\cdot|s)\right)\right].$$

The complete proof can be found in Appendix A.1. Since $q_{\pi^*}(s,a)$ *represents the expected return under the decent policy $\pi^*$ when staring from $(s,a)$ and reaches the maximum if $\pi^*$ is truly optimal, it can be interpreted as the importance of $(s,a)$*, and higher return is likely to be achieved when starting from more important $(s,a)$. Consequently, the gap between $J(\pi^*)$ and $J(\pi)$ directly depends on the importance-weighted decision difference between $\pi^*$ and $\pi$. Based on Theorem 2 and $q_{\pi^*} \ge 0$, we can readily derive a bound on the guarantee on $|J(\pi^*) - J(\pi)|$ by applying the triangle inequality.

**Corollary 1.** *Given two policies of $\pi^*$ and $\pi$ with $\mathbb{E}_{s\sim d_\pi(s)}\left[q_{\pi^*}(s,\cdot)\left|\pi^*(\cdot|s)-\pi(\cdot|s)\right|\right] \le \epsilon$, we have $|J(\pi^*) - J(\pi)| \le \frac{1}{1-\gamma}\epsilon$.*

**Tightness** Since only the triangle inequality is applied, there exists the worst case for $\pi$ where $\pi^*(a|s) - \pi(a|s) < 0$ holds only when $q_{\pi^*}(s,a) = 0$. This makes the inequality collapse to equality in Corollary 1, thereby demonstrating that the upper bound in Corollary 1 is *non-vacuous*.

**Algorithm 1:** Action-value weighted PBC

---

**Input** : offline RL dataset $\mathcal{D}_{\texttt{off}}$, synthetic data size $N_{\texttt{syn}}$, loop step $T$, $T_{\texttt{out}}$, learning rate $\alpha_0$, $\alpha_1$, momentum rate $\beta_0$, $\beta_1$

**Output** : synthetic dataset $\mathcal{D}_{\texttt{syn}}$

$\pi^*, q_{\pi^*} \leftarrow \texttt{OfflineRL}(\mathcal{D}_{\texttt{off}})$

Initialize $\mathcal{D}_{\texttt{syn}} = \{(s_i, a_i)\}_{i=1}^{N_{\texttt{syn}}}$ by randomly sampling $(s_i, a_i) \sim \mathcal{D}_{\texttt{off}}$

**for** $t_{out} = 1$ **to** $T_{out}$ **do**

    Randomly initialize policy network parameters $\theta_0$

    ▷ Behavioral cloning with synthetic data.

    **for** $t = 1$ **to** $T$ **do**

        Compute the BC loss *w.r.t.* synthetic data $\mathcal{L}_{t-1} = \ell_{\texttt{BC}}(\theta_{t-1}, \mathcal{D}_{\texttt{syn}})$

        Update $\theta_t \leftarrow \texttt{GradDescent}(\nabla_{\theta_{t-1}} \mathcal{L}_{t-1}, \alpha_0, \beta_0)$

    **end**

    Construct the minibatch $\mathcal{B} = \{(s_i, a_i)\}_{i=1}^{|\mathcal{B}|}$ by sampling $s_i \sim \mathcal{D}_{\texttt{off}}$ and $a_i \sim \pi^*(\cdot|s_i)$

    Compute $\mathcal{H}(\pi_{\theta_T}, \mathcal{B}) = \frac{1}{|\mathcal{B}|} \sum_{i=1}^{|\mathcal{B}|} q_{\pi^*}(s_i, a_i) (\pi_{\theta_T}(a_i|s_i) - \pi^*(a_i|s_i))^2$

    Update $\mathcal{D}_{\texttt{syn}} \leftarrow \texttt{GradDescent}(\nabla_{\mathcal{D}_{\texttt{syn}}} \mathcal{H}(\pi_{\theta_T}, \mathcal{B}), \alpha_1, \beta_1)$

**end**

---

**Comparison to Thm. 1** With the fact $q_{\pi^*}(s, a) \leq \sum_{t=0}^{\infty} R_{\max} = \frac{R_{\max}}{1-\gamma}$, we have

$$\mathbb{E}_{s \sim d_\pi(s)} [q_{\pi^*}(s, \cdot) |\pi^*(\cdot|s) - \pi(\cdot|s)|] \leq \frac{R_{\max}}{1-\gamma} \mathbb{E}_{s \sim d_\pi(s)} [|\pi^*(\cdot|s) - \pi(\cdot|s)|], \tag{5}$$

therefore our bound in Corollary 1 is significantly tighter than Theorem 1, as $q_{\pi^*}(s, a) = \sum_{t=0}^{\infty} R_{\max}$ requires $\pi^*$ to achieve the maximum reward at every step. This condition is particularly difficult for sparse-reward environments where most $r(s, a)$ are close to zero. Moreover, combining the proof of Theorem 2 and Eq. 5 provides a more straightforward proof of Theorem 1.

As shown by the theoretical analysis, action-value weighted objective offers stronger distillation guarantees due to the linear discount factor complexity $\mathcal{O}(1/(1-\gamma))$. This improvement alleviates the loose guarantee caused by limited optimization in OBD compared to former quadratic $\mathcal{O}(1/(1-\gamma)^2)$. Accordingly, we propose **action-value weighted PBC (Av-PBC)** as the OBD objective:

$$\mathcal{H}(\pi, \mathcal{D}_{\texttt{off}}) = \mathbb{E}_{s \sim \mathcal{D}_{\texttt{off}}} \left[ q_{\pi^*}(s, \cdot) (\pi(\cdot|s) - \pi^*(\cdot|s))^2 \right]. \tag{6}$$

While Av-PBC is theoretically induced, it is quite intuitive to understand: states $s$ in $\mathcal{D}_{\texttt{off}}$ are normally sampled by a mixture of policies instead of the expert $\pi^*$. If we sampled a bad state $s$ with extremely small $q_{\pi^*}(s, a)$, measuring the decision difference between $\pi$ and $\pi^*$ will be less important. As for practical implementation, Eq. 6 requires summing over the entire action space $\mathcal{A}$ to compute $\sum_{a \in \mathcal{A}}$, which is highly inefficient for large $|\mathcal{A}|$. Considering the expert policy is typically highly concentrated, *i.e.*, only a few actions are selected by $\pi^*$ with large action values, we instead sample $a \sim \pi^*(\cdot|s)$ to efficiently estimate Eq. 6. The pseudo-code of Av-PBC is presented in Algorithm 1.

## 5 Experiments

In this section, we evaluate the proposed ODB algorithms across multiple offline RL datasets from perspectives of (1) distillation performance, (2) distillation convergence speed, (3) cross-architecture and cross-optimizer generalization, and (4) policy ensemble performance *w.r.t.* distilled data.

**Datasets** We conduct offline behavior distillation on D4RL [Fu et al., 2020], a widely used offline RL benchmark. Specifically, OBD algorithms are evaluated on three popular environments of `Halfcheetah`, `Hopper`, and `Walker2D`. For each environment, three offline RL datasets of varying quality are provided by D4RL, *i.e.*, `medium-replay` (M-R), `medium` (M), and `medium-expert` (M-E) datasets. Thus, a total of $3 \times 3 = 9$ datasets are employed to assess OBD algorithms. `medium` dataset is collected from the environment with "medium" level policies; `medium-replay` dataset consists of recording all samples in the replay buffer observed during training this "medium" level policy; and `medium-expert` dataset is a mixture of expert demonstrations and sub-optimal data.

Table 1: Offline behavior distillation performance on D4RL offline datasets. The result for Random Selection (Random) is obtained by repeating 10 times. For DBC, PBC, and Av-PBC, the results are averaged across five seeds and the last five evaluation steps. The best OBD result for each dataset is marked with **bold** scores, and orange-colored scores denote instances where OBD outperforms BC.

| Method | Halfcheetach | | | Hopper | | | Walker2D | | | Average |
|--------|------|------|------|------|------|------|------|------|------|---------|
| | M-R | M | M-E | M-R | M | M-E | M-R | M | M-E | |
| Random | 0.9 | 1.8 | 2.0 | 19.1 | 19.2 | 11.6 | 1.9 | 4.9 | 6.7 | 7.6 |
| DBC | 2.5 | 28.2 | **29.0** | 12.1 | **37.8** | 31.1 | 6.1 | 29.3 | 11.7 | 20.9 |
| PBC | 19.4 | 30.9 | 20.5 | 35.6 | 25.1 | 33.4 | 41.5 | 33.2 | 34.0 | 30.4 |
| Av-PBC | **35.9** | **36.9** | 22.0 | **40.9** | 32.5 | **38.7** | **55.0** | **39.5** | **42.1** | **38.2** |
| BC (Whole) | 14.0 | 42.3 | 59.8 | 22.9 | 50.2 | 51.7 | 14.6 | 65.9 | 89.6 | 45.7 |
| OffRL (Whole) | 45.8 | 47.6 | 50.8 | 98.0 | 56.4 | 107.3 | 87.4 | 84.0 | 109.0 | 70.1 |

**Setup** The advanced offline RL algorithm of Cal-QL [Nakamoto et al., 2023] is utilized to extract the decent $\pi^*$ and $q_{\pi^*}$ from $\mathcal{D}_{\texttt{off}}$. A four-layer MLP serves as the default architecture for policy networks. The size of synthetic data $N_{\texttt{syn}}$ is set to 256. Standard SGD is employed in both inner and outer optimization, and learning rates $\alpha_0 = 0.1$ and $\alpha_1 = 0.1$ for the inner and outer loop, respectively, and corresponding momentum rates $\beta_0 = 0$ and $\beta_1 = 0.9$. Additional implementation details are provided in Appendix B.

**Evaluation** To accesss the performance of $\mathcal{D}_{\texttt{syn}}$, we train policy networks on $\mathcal{D}_{\texttt{syn}}$ with standard BC, and obtain the corresponding averaged return by interacting with the environment for 10 episodes. We use `normalized return` [Fu et al., 2020] for better visualization: `normalized return` $= 100 \times \frac{\texttt{return - random return}}{\texttt{expert return - random return}}$, where `random return` and `expert return` refer to returns of random policies and the expert policy (online SAC [Haarnoja et al., 2018]), respectively.

**Baselines** (1) *Random Selection*: randomly selecting $N_{\texttt{syn}}$ real state-action pairs from $\mathcal{D}_{\texttt{off}}$; (2) *DBC*; (3) *PBC*; (4) *Av-PBC*. We also report policy performance of behavioral cloning and Cal-QL in terms of training on the whole offline dataset $\mathcal{D}_{\texttt{off}}$ for a comprehensive comparison.

## 5.1 Main Results

We first investigate the performance of various OBD algorithms (DBC, PBC, Av-PBC) across offline datasets of varying quality and environments, as detailed in Table 1. Several observations are obtained from the results: (1) offline behavior distillation effectively synthesize informative data that enhance policy training (DBC/PBC/Av-PBC *vs.* Random Selection); (2) PBC demonstrates better distillation performance than the basic DBC, especially given the low-quality RL data, highlighting the benefit of action correction in the sub-optimal data (30.4 *vs.* 20.9); (3) Av-PBC considerably outperforms PBC across all datasets (38.2 *vs.* 30.4); (4) when the offline data are collected by low-quality policies (`medium-replay`), Av-PBC can surpass BC trained on the whole data, while it gradually lags behind BC with higher-quality offline data (`medium-replay` and `medium-expert`); (5) given that the objective of OBD is to approximate the decent policy extracted by offline RL algorithms, offline RL serves as an upper bound for OBD performance. In summary, the empirical results show that Av-PBC increases OBD performance by a substantial margin compared to the baselines (82.8% for DBC and 25.7% for PBC).

An interesting phenomenon observed with Av-PBC is that *synthetic data distilled from* `medium-replay` *offline datasets exhibit better performance than those distilled from* `medium-expert` *offline datasets*. We explain here: while `medium-expert` data offer better quality, `medium-replay` data contains more diverse states due to being sampled by a mixture of less-trained policies that explore a wider rage of states. This is similar to exploration-exploitation dilemma in RL [Sutton and Barto, 2018] and underscores the importance of state coverage in original data for OBD.

**Training Time Comparison** To further illustrate the advantages of OBD, we compare the time required for training polices on original data versus OBD-distilled data. For synthetic data with a size of 256, only 100 optimization steps are necessary, corresponding to a training time of 0.2s, while 25k~125k steps are required for BC on original data. With distilled data, the training time can be reduced by over 99.5%. A detailed list of training steps for all datasets is provided in Appendix C.

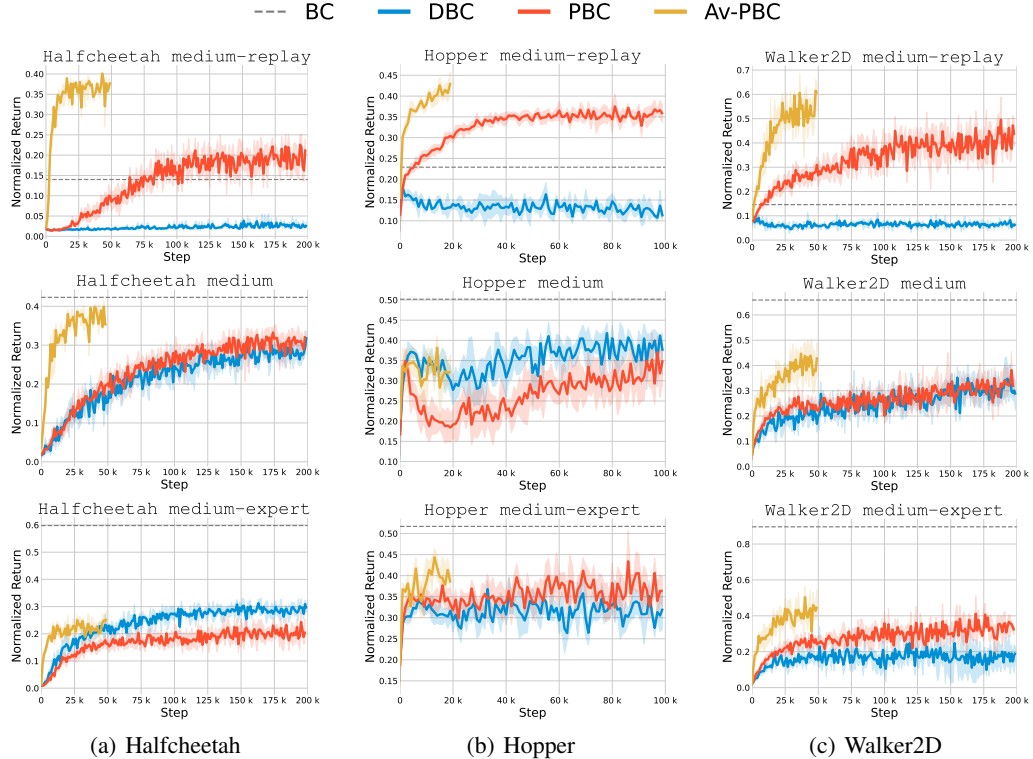

| | | | | |
|---|---|---|---|---|
| | (a) Halfcheetah | (b) Hopper | (c) Walker2D | |

Figure 1: Plots of OBD performance, represented by the normalized returns of policies trained on synthetic data, as functions of distillation steps on (a) Halfcheetah; (b) Hopper; and (c) Walker2D environment. Each curve is averaged over five random seeds.

Table 2: Offline behavior distillation performance across various policy network architectures and optimizers (Optim). Red-colored scores and green-colored scores in brackets denote the performance degradation and improvement, respectively, compared to the default training setting. The results are averaged over five random seeds and the last five evaluation steps.

| | Arch/Opt | Halfcheetach | | | Hopper | | | Walker2D | | | Average |
|---|---|---|---|---|---|---|---|---|---|---|---|
| | | M-R | M | M-E | M-R | M | M-E | M-R | M | M-E | |
| Architecture | 2-layer | 37.1 | 35.9 | 10.9 | 29.9 | 26.2 | 33.9 | 49.2 | 41.3 | 51.1 | 35.1 (3.1) |
| | 3-layer | 38.6 | 39.7 | 19.4 | 39.0 | 28.1 | 41.5 | 63.2 | 44.1 | 55.3 | 41.0 (2.8) |
| | 5-layer | 36.1 | 37.7 | 20.0 | 37.1 | 29.1 | 36.6 | 52.0 | 36.7 | 31.6 | 35.2 (3.0) |
| | 6-layer | 32.1 | 36.0 | 17.3 | 36.9 | 29.6 | 32.8 | 47.1 | 28.2 | 25.5 | 31.7 (6.5) |
| | Residual | 36.9 | 36.4 | 20.0 | 38.8 | 29.8 | 40.3 | 47.5 | 35.7 | 37.1 | 35.8 (2.4) |
| Optim | Adam | 35.8 | 37.6 | 22.9 | 40.5 | 31.2 | 40.2 | 55.8 | 41.9 | 47.7 | 39.3 (1.1) |
| | AdamW | 36.8 | 37.9 | 21.4 | 40.6 | 33.3 | 41.1 | 55.4 | 44.2 | 43.2 | 39.3 (1.1) |
| | SGDm | 36.4 | 37.3 | 21.8 | 40.4 | 30.9 | 39.2 | 54.7 | 40.2 | 42.1 | 38.1 (0.1) |

**Convergence Speed of OBD** To compare the convergence speed of OBD algorithms, we plot the performance of various OBD algorithms over distillation step; please see Figure 1. These plots demonstrate that Av-PBC not only improves the OBD performance, but has significant convergence speed and requires only a quarter of the distillation steps compared to DBC and PBC, which is essential for OBD considering the compute-intensive bi-level optimization.

**Cross Architecture and Optimizer Performance** We evaluate the synthetic data across various training configurations to assess the cross-architecture/optimizer generalization of Av-PBC. Concretely, we employ the data distilled by Av-PBC with the default network (4-layer MLP) and optimizer (SGD) to train (1) different networks of 2/3/5/6-layer and residual MLPs and (2) the default 4-layer MLP with different optimizers of Adam, AdamW, and SGDm (SGD with momentum=0.9). The

Table 3: Offline behavior distillation performance on D4RL offline datasets with ensemble num of 10. Green-colored scores in brackets denote the performance improvement compared to the non-ensemble setting. The results are averaged over five random seeds and the last five evaluation steps.

| Method | Halfcheetach | | | Hopper | | | Walker2D | | | Average |
|--------|------|------|------|------|------|------|------|------|------|---------|
| | M-R | M | M-E | M-R | M | M-E | M-R | M | M-E | |
| DBC | 2.0 | 30.0 | 31.8 | 9.3 | 44.9 | 43.3 | 5.8 | 50.6 | 33.6 | 27.9 (7.0) |
| PBC | 12.9 | 33.4 | 31.6 | 36.6 | 36.7 | 41.8 | 64.1 | 41.6 | 42.0 | 37.9 (7.3) |
| Av-PBC | 39.8 | 41.4 | 37.2 | 39.7 | 27.6 | 38.8 | 75.9 | 58.6 | 73.7 | 48.1 (9.9) |

results are presented in Table 2. As shown in the last column of average performance, we observe that (1) albeit a slight drop, synthetic data distilled by Av-PBC are still valid in training different policy networks, and (2) the performance of distilled data is relatively robust to the variation of optimizers. Therefore, the Av-PBC-distilled data possess satisfied cross-architecture/optimizer performance.

**Policy Ensemble on OBD Data**  With the tiny distilled dataset, policy ensemble can be efficiently performed to further enhance policy performance. This is achieved by training multiple policy networks on synthetic data and then combining their outputs to generate actions. To evaluate the performance gain from policy ensemble, we train 10 policy networks with different seeds; please see Table 3. The results demonstrate that (1) policies trained on synthetic data can be substantially boosted through ensemble (25.8% for Av-PBC); and (2) Av-PBC exhibits a larger performance gain than DBC and PBC (9.9 *vs.* 7.0/7.3), highlighting the advantages of Av-PBC in policy ensemble.

# 6    Discussion

**Applications**  Distilled behavioral data encapsulate critical decision-making knowledge from offline RL data and associated environment, making them highly applicable to various downstream RL tasks. Through BC on distilled data, we can *rapidly pretrain a good policy* for online RL fine-tuning [Goecks et al., 2019]. On the other hand, after offline pretraining, the policy can be further enhanced by online fine-tuning, while there exists *catastrophic forgetting w.r.t.* offline data knowledge during fine-tuning [Luo et al., 2023]. To tackle this challenge, Zhao et al. [2022] propose to use BC loss *w.r.t.* offline data as a constraint during the fine-tuning phase. By replacing the massive offline data with distilled data, we can achieve more efficient loss computation and thus better algorithm convergence. A similar approach can be achieved to circumvent catastrophic forgetting in continual offline RL [Gai et al., 2023], where the goal is to learn a sequence of offline RL tasks while retaining good performance across all tasks. Moreover, *multi-task offline RL* [Yu et al., 2021], which learns multiple RL tasks jointly from a combination of specific offline datasets, also receives benefits from OBD in terms of efficiency by alternative training on the mixture of distilled data via BC [Lupu et al., 2024].

Beyond benefits in efficient policy training, OBD shows potential for *protecting data privacy*: given that offline datasets often contain sensitive information, such as medical records, privacy concerns are significant in offline RL due to various privacy attacks on the learned policies [Qiao and Wang, 2023]. OBD can enhance privacy preservation by publishing smaller, distilled datasets instead of the full, sensitive data. Besides, distilled behavioral data is also beneficial for *explainable RL* by highlighting the critical states and corresponding actions. A example of this is provided in Appendix D.

**Limitations**  The OBD data are 2-tuples of (`state`, `action`) and exclude `reward`. Thus, the distilled data are solely leveraged by the supervised BC and invalid for conventional RL algorithms with Bellman backup. Despite this deficiency, OBD data can still facilitate the applications above by efficiently injecting high-quality decision-making knowledge into policy networks with BC loss.

We note that two major challenges remain in current OBD algorithms: distillation inefficiency and policy performance degradation. While our Av-PBC substantially decreases the distillation steps, the OBD process is still computationally expensive (25 hours for 50k distillation steps on a single NVIDIA V100 GPU) due to the bi-level optimization involved. Moreover, there remains a notable performance gap between OBD and the whole data with offline RL algorithms (38.2 *vs.* 70.1 in Table 1). These limitations also shed light on future directions in improving the efficiency of OBD and bridging the gap between synthetic data and the original offline RL dataset.

# 7 Conclusion

In this paper we integrate the advanced dataset distillation with offline RL data, formalizing the concept of offline behavior distillation (OBD). We introduce two OBD objectives: the naive offline data-based BC (DBC) and its policy-corrected variant, PBC. Through comprehensive theoretical analysis, we demonstrate that PBC offers inferior OBD performance guarantee of $\mathcal{O}(1/(1-\gamma)^2)$ under complex bi-level optimization, which inevitably incurs significant distillation loss.. To tackle this issue, we theoretically establish the equivalence between policy performance gap and action-value weighted decision difference, leading to the proposal of action-value weighted BC (Av-PBC). This novel Av-PBC objective significantly improves the performance guarantee to $\mathcal{O}(1/(1-\gamma))$. Extensive experiments on multiple offline RL datasets demonstrate that Av-PBC vastly enhances OBD performance and accelerates the distillation process by several times.

## Acknowledge

The authors thank the anonymous reviewers for their helpful comments and feedback. The authors are also grateful to Zhihao Cheng for thoughtful discussions and fruitful comments. Dr Tao is partially supported by NTU RSR and Start Up Grants.

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

# A  Proofs

## A.1  Proof of Theorem 2

*Proof.* With the definitions $\rho_\pi^n(s,a) = \pi(a|s)d_\pi^{n-1}(s)$ and $J(\pi) = \mathbb{E}_{\rho_\pi^1(s,a)}\left[q_\pi(s,a)\right]$, we have

$$
\begin{aligned}
&J(\pi^*) - J(\pi)\\
&= \mathbb{E}_{\rho_{\pi^*}^1(s,a)}\left[q_{\pi^*}(s,a)\right] - \mathbb{E}_{\rho_\pi^1(s,a)}\left[q_\pi(s,a)\right]\\
&= \sum_{(s,a)\in\mathcal{S}\times\mathcal{A}}\left[\rho_{\pi^*}^1(s,a)q_{\pi^*}(s,a) - \rho_\pi^1(s,a)q_\pi(s,a)\right]\\
&= \sum_{(s,a)\in\mathcal{S}\times\mathcal{A}}\left[\pi^*(a|s)d_{\pi^*}^0(s)q_{\pi^*}(s,a) - \pi(a|s)d_\pi^0(s)q_\pi(s,a)\right]\\
&= \sum_{(s,a)\in\mathcal{S}\times\mathcal{A}}\Big[\pi^*(a|s)d_{\pi^*}^0(s)q_{\pi^*}(s,a) - \pi(a|s)d_{\pi^*}^0(s)q_{\pi^*}(s,a)\\
&\qquad + \pi(a|s)d_{\pi^*}^0(s)q_{\pi^*}(s,a) - \pi(a|s)d_\pi^0(s)q_\pi(s,a)\Big]\quad (d_{\pi^*}^0(s)\equiv d_\pi^0(s)\equiv d^0(s))\\
&= \mathbb{E}_{s\sim d_{\pi^*}^0(s)}\left[\sum_{a\in\mathcal{A}}\left(\pi^*(a|s) - \pi(a|s)\right)q_{\pi^*}(s,a)\right] + \mathbb{E}_{\rho_\pi^1(s,a)}\left[q_{\pi^*}(s,a) - q_\pi(s,a)\right]\quad (7)
\end{aligned}
$$

For the term $q_{\pi^*}(s,a) - q_\pi(s,a)$, we have

$$
\begin{aligned}
&q_{\pi^*}(s,a) - q_\pi(s,a)\\
&= r(s,a) + \gamma\mathbb{E}_{s'\sim\mathcal{T}(s'|s,a)}\left[\sum_{a'\in\mathcal{A}}\pi^*(a'|s')q_{\pi^*}(s',a')\right]\\
&\quad - r(s,a) - \gamma\mathbb{E}_{s'\sim\mathcal{T}(s'|s,a)}\left[\sum_{a'\in\mathcal{A}}\pi(a'|s')q_\pi(s',a')\right]\\
&= \gamma\mathbb{E}_{s'\sim\mathcal{T}(s'|s,a)}\left[\sum_{a'\in\mathcal{A}}\pi^*(a'|s')q_{\pi^*}(s',a') - \pi(a'|s')q_\pi(s',a')\right]\quad (8)
\end{aligned}
$$

Furthermore, due to $d_\pi^{n+1}(s') = \rho_\pi^n(s,a)\mathcal{T}(s'|s,a)$ we have

$$
\begin{aligned}
&\mathbb{E}_{\rho_\pi^n(s,a)}\left[q_{\pi^*}(s,a) - q_\pi(s,a)\right]\\
&= \gamma\mathbb{E}_{\rho_\pi^n(s,a)}\left[\mathbb{E}_{s'\sim\mathcal{T}(s'|s,a)}\left[\sum_{a'\in\mathcal{A}}\pi^*(a'|s')q_{\pi^*}(s',a') - \pi(a'|s')q_\pi(s',a')\right]\right]\\
&= \gamma\mathbb{E}_{s\sim d_\pi^{n+1}(s)}\left[\sum_{a\in\mathcal{A}}\pi^*(a|s)q_{\pi^*}(s,a) - \pi(a|s)q_\pi(s,a)\right]\\
&= \gamma\mathbb{E}_{s\sim d_\pi^{n+1}(s)}\left[\sum_{a\in\mathcal{A}}\pi^*(a|s)q_{\pi^*}(s,a) - \pi(a|s)q_{\pi^*}(s,a) + \pi(a|s)q_{\pi^*}(s,a) - \pi(a|s)q_\pi(s,a)\right]\\
&= \gamma\mathbb{E}_{s\sim d_\pi^{n+1}(s)}\left[\sum_{a\in\mathcal{A}}\left(\pi^*(a|s) - \pi(a|s)\right)q_{\pi^*}(s,a)\right] + \gamma\mathbb{E}_{s\sim d_\pi^{n+1}(s)}\left[\sum_{a\in\mathcal{A}}\pi(a|s)q_{\pi^*}(s,a) - \pi(a|s)q_\pi(s,a)\right]\\
&= \gamma\mathbb{E}_{s\sim d_\pi^{n+1}(s)}\left[\sum_{a\in\mathcal{A}}\left(\pi^*(a|s) - \pi(a|s)\right)q_{\pi^*}(s,a)\right] + \gamma\mathbb{E}_{\rho_\pi^{n+1}(s,a)}\left[q_{\pi^*}(s,a) - q_\pi(s,a)\right]\quad (9)
\end{aligned}
$$

Plugging the iterative formula of Eq. 9 into Eq. 7 yields the desired equality:

$$
\begin{aligned}
& J(\pi^*) - J(\pi) \\
&= \mathbb{E}_{s \sim d_{\pi^*}^0(s)} \left[ \sum_{a \in \mathcal{A}} \left( \pi^*(a|s) - \pi(a|s) \right) q_{\pi^*}(s, a) \right] + \mathbb{E}_{\rho_{\pi}^1(s,a)} \left[ q_{\pi^*}(s, a) - q_{\pi}(s, a) \right] \\
&= \sum_{t=0}^{\infty} \gamma^t \mathbb{E}_{s \sim d_{\pi}^t(s)} \left[ \sum_{a \in \mathcal{A}} \left( \pi^*(a|s) - \pi(a|s) \right) q_{\pi^*}(s, a) \right] \\
&= \frac{1}{1 - \gamma} \mathbb{E}_{s \sim d_{\pi}(s)} \left[ \sum_{a \in \mathcal{A}} \left( \pi^*(a|s) - \pi(a|s) \right) q_{\pi^*}(s, a) \right]
\end{aligned}
\tag{10}
$$

The last equation uses the definition that $d_\pi(s) = (1-\gamma) \sum_{t=0}^{\infty} \gamma^t d_\pi^t(s)$. The proof is completed. $\square$

## B  Implementation Details

This section provides all the additional implementation details of our experiments.

**OBD Settings**  The policy network is a 4-layer multilayer perceptron (MLP) with a width of 256. The synthetic data are initialized by randomly selecting $N_{\text{syn}}$ state-action pairs from the offline data. For DBC and PBC, the distillation step $T_{\text{out}}$ is set to 200k for `Halfcheetah` and `Walker2D` and 50k for `Hopper`, respectively. For Av-PBC, the distillation step $T_{\text{out}}$ is set to 50k for `Halfcheetah` and `Walker2D` and 20k for `Hopper`, respectively. The inner loop step $T_{\text{in}}$ is set to 100.

**Offline RL Policy Training**  We use the advanced offline RL algorithm of Cal-QL [Nakamoto et al., 2023] to extract the decent policy $\pi^*$ and corresponding q value function $q_{\pi^*}$ from sub-optimal offline data, and the implementation in [Tarasov et al., 2022] is employed in our experiments with default hyper-parameter setting.

**Cross-architecture Experiments.**  The width of MLPs are both 256. The residual MLP is a 4-layer MLP, and the intermediate layers are packaged into the residual block.

## C  Training Time Comparison

Table 4: The size and required training steps for convergence for each offline dataset. M denotes the million for simplicity. The size and step for synthetic data (Synset) are listed in the last column.

| | Halfcheetach | | | Hopper | | | Walker2D | | | Synset |
|---|---|---|---|---|---|---|---|---|---|---|
| | M-R | M | M-E | M-R | M | M-E | M-R | M | M-E | |
| Size | 0.2M | 1M | 2M | 0.4M | 1M | 2M | 0.3M | 1M | 2M | 256 |
| Step (k) | 40 | 25 | 100 | 80 | 50 | 100 | 60 | 50 | 125 | 0.1 |

For the whole original data, offline RL algorithms require dozens of hours. Therefore, we solely compare the training time of BC on synthetic data and BC on original data. Because the training time varies with GPU models (NVIDIA V100 used in our experiments), we report the optimization step, which has a linear relationship to training time, required for training convergence for each original dataset, as shown in Table 4.

## D    Examples of Distilled Data

We present several examples of distilled behavioral data for Halfcheetah in Figure 2. The top row illustrates the distilled states, while the bottom row depicts the subsequent states after executing the corresponding distilled actions within the environment. The figure demonstrates that (1) the distilled states prioritize "critical states" or "imbalanced states" (for the cheetah) more than "balanced states"; and (2) the states following the execution of distilled actions are closer to "balanced states" compared to the initial distilled states. These examples offer insights into the explainability of reinforcement learning processes.

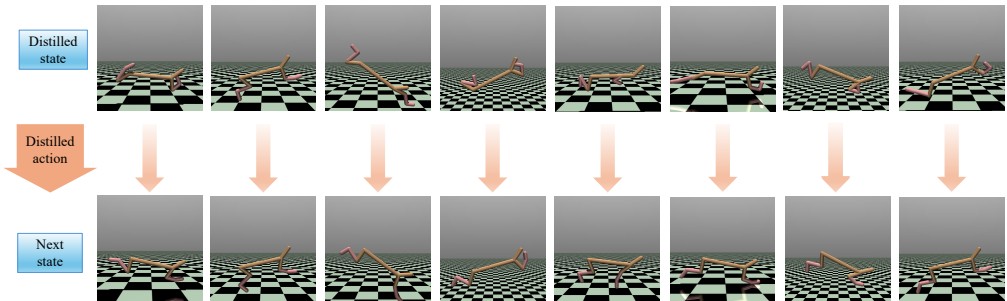

Figure 2: Examples of distilled behavioral data. The top row shows the distilled states, while the bottom row presents the subsequent state following the execution of the corresponding distilled actions within the environment.

## E    The Performance of Av-PBC across Different Synthetic Data Sizes

We investigate the impact of varying synthetic data size on OBD performance. The results, as shown in Table 5, suggest that OBD performance improves with an increase in synthetic data size. This enhancement is attributed to the larger synthetic datasets conveying more comprehensive information regarding the RL environment and associated decision-making processes.

Table 5: The Av-PBC performance on D4RL offline datasets with different synthetic data sizes.

| Dataset | Synthetic Data Size | | | | |
|---|---|---|---|---|---|
| | **16** | **32** | **64** | **128** | **256** |
| Halfcheetah M-R | 6.9 | 15.3 | 23.8 | 33.2 | 35.9 |
| Hopper M-R | 27.3 | 29.9 | 32.3 | 38.1 | 40.9 |
| Walker2D M-R | 14.8 | 21.8 | 34.0 | 50.0 | 55.0 |

