# OpenReview forum: "Offline Behavior Distillation"
_NeurIPS.cc/2024/Conference — NeurIPS 2024 poster_

### Official Review · Reviewer_2J9f · 2024-07-10

**Soundness:** 3
**Presentation:** 2
**Contribution:** 2
**Rating:** 5
**Confidence:** 4

**Summary:**

This paper emphasizes on the offline data generation to enable rapid policy learning in reinforcement learning. A new surrogate loss function is proposed for the data generation. Theoretical analysis shows the superiority of the proposed method in the performance gap.

**Strengths:**

1. A new objective for data generation is proposed with a performance guarantee.
2. Experiments are conducted over multiple benchmark datasets.

**Weaknesses:**

1. Related work on offline data generation of RL should be provided and compared. The proposed Av-PBC method is only compared with random policy, are there other related works? Is this paper the first work of offline data generation in RL?
2. In section 1, it is not clear what is the difference between AvPBC and Av-PBC. Only Av-PBC is evaluated in the experiments.
3. The network architecture is not clear. Since data generation is required. A generating network should be used in addition to a policy network. More details about the data generation model should be provided.
4. In the experiments, what is the number of offline datasets as the number of offline data samples influences performance?
5. In the experiments, the time and performance of Av-PBC and OffRL should be compared under varying numbers of generated data.

**Questions:**

Please refer to Weakness.

---

> ### Author Rebuttal · Authors · 2024-08-05
>
> **Q1:** *Related work on offline data generation of RL should be provided and compared. The proposed Av-PBC method is only compared with random policy, are there other related works? Is this paper the first work of offline data generation in RL?*
>
> **A1:** Thanks for your comments. To the best of our knowledge, our paper is the first to achieve behavioral data distillation from offline RL data.  We introduced the novel OBD algorithm Av-PBC and demonstrated its effectiveness by comparing it with two baseline methods, DBC and PBC. Our method shows significant improvements in both performance and speed across multiple datasets.
>
> The most closely related work to ours is (online) behavioral distillation [1]. We have detailed the differences between [1] and our work in the Related Works section (Line 103-114). Our focus is on the offline scenario, where the RL environment is not accessible. This distinction is crucial as it highlights the unique challenges and contributions of our approach; please refer to Line 103-114 for more details.
>
> &nbsp;
>
> [1] Andrei Lupu, Chris Lu, Jarek Luca Liesen, Robert Tjarko Lange, and Jakob Nicolaus Foerster. Behaviour distillation. In *The Twelfth International Conference on Learning Representations*, 2024.
>
>
> &nbsp;
>
>
> **Q2:** *It is not clear what is the difference between AvPBC and Av-PBC in Section 1.*
>
> **A2:** Thanks and addressed. "AvPBC" is "Av-PBC" in Section 1.
>
> &nbsp;
>
> **Q3:** *The network architecture is not clear. Since data generation is required. A generating network should be used in addition to a policy network. More details about the data generation model should be provided.*
>
> **A3:** Thanks for your suggestion. We would like to clarify that our OBD method does not involve the use of generative models. Instead, we optimize the distilled data through backpropagation, applying an element-wise update as outlined by the $\texttt{GradDescent}$ procedure in Algorithm 1.
>
>
> In the first step of $\texttt{GradDescent}$, we calculate the meta-gradient $\nabla_\mathcal{D_\textbf{syn}}$ via Backpropagation Through Time (BPTT) as described in Equation 3. We then update $\mathcal{D_\textbf{syn}}$ by $\mathcal{D_\textbf{syn}} \rightarrow \mathcal{D_\textbf{syn}} - \alpha \nabla_\mathcal{D_\textbf{syn}} $. We will provide additional clarification and detail about this process in our revised paper.
>
> &nbsp;
>
> **Q4:** *What is the number of offline datasets as the number of offline data samples influences performance?*
>
> **A4:** Thank you for your comments. We list the sample size of each offline dataset in Table 3 of the PDF file, which can also be found in Table 4 of the Appendix. These datasets range from 0.3 million to 2 million samples. Our Av-PBC significantly outperforms other approaches across these varying dataset sizes, demonstrating its robustness and effectiveness.
>
> &nbsp;
>
> **Q5:** *The time and performance of Av-PBC and OffRL should be compared under varying numbers of generated data.*
>
> **A5:** Thanks for your suggestion. We explored the effect of different sizes of synthetic datasets on OBD performance during the response period. The results, presented in Table 2 of the PDF file, indicate that as the size of the synthetic data increases, OBD performance improves. This enhancement is attributed to the larger synthetic datasets conveying more comprehensive information about the RL environment and decision-making processes.

---

> > ### Comment · Reviewer_2J9f · 2024-08-09
> >
> > Thank you for your response. I will maintain my current score.

---

### Official Review · Reviewer_VAdy · 2024-07-12

**Soundness:** 3
**Presentation:** 3
**Contribution:** 3
**Rating:** 6
**Confidence:** 3

**Summary:**

The paper considers the offline behavior distillation (OBD) for reinforcement learning. The problem is to distill a synthetic dataset given a large dataset sampled by a sub-optimal policy.  The key challenge of the problem is to design a good distillation objective. The authors first give two native objectives and prove they are limited to provide good performance guarantee. Then, they propose the Action-value weighted PBC objective and prove a better performance guarantee than the naive objectives. The authors evaluate the proposed algorithms on the dataset D4RL and demonstrate the distillation performance and show the advantage of OBD in reducing the RL training time.

**Strengths:**

The OBD problem is well motivated and closely related to the concerns of the community.

**Weaknesses:**

- Corollary 1 and Corollary 2 compare the policy for dataset distillation to the policy of offline RL, but we are more concerned about the performance of any policy trained on the distilled dataset comparing to the optimal policy. It would be better to propose a new metric to evaluate the performance of OBD for RL.

**Questions:**

See the weakness section.  Also, I have the following comments.

1. It is also a challenge to decide the size of the synthetic dataset to strike a balance between training efficiency and performance.
2. I am also  concerned about the scaling issue. If the synthetic dataset  is large, the dataset distillation has a high computational cost.

**Limitations:**

The authors adequately addressed the limitations.

---

> ### Author Rebuttal · Authors · 2024-08-05
>
> **Q1:** *Corollary 1 and Corollary 2 compare the policy for dataset distillation to the policy of offline RL, but we are more concerned about the performance of any policy trained on the distilled dataset comparing to the optimal policy. It would be better to propose a new metric to evaluate the performance of OBD for RL.*
>
> **A1:** Thanks for your comments. As there is no assumption w.r.t. policies, we would like to clarify that in Corollaries 1 and 2, the policy $\pi^\ast$ can represent either a decent policy learned from an offline RL algorithm or the optimal policy. Since the environment and optimal policy are not accessible in the offline setting, we use advanced offline RL algorithms to derive an "estimated expert policy" $\pi^\ast$ from the offline RL data for the loss computation of OBD. Corollaries 1 and 2 theoretically show that this approach effectively assesses the performance of policies trained on the distilled dataset in relation to the practical benchmark policy $\pi^\ast$, given the constraints of the offline RL scenario.
>
> &nbsp;
>
>
> **Q2:**  *It is also a challenge to decide the size of the synthetic dataset to strike a balance between training efficiency and performance.*
>
> **A2:** Thanks for your comments. We have tested various sizes of synthetic datasets, and the results are presented in Table 1 of the PDF. This table demonstrates that OBD performance improves with larger synthetic datasets, as they provide more comprehensive RL environment and decision information during the OBD process.
>
> However, a larger synthetic dataset also presents challenges: (1) it increases the number of parameters in the bi-level optimization process, raising computational costs; and (2) it can reduce efficiency when training policies on the synthetic data. Determining the optimal size of the synthetic dataset depends on the specific RL environment and the coverage and quality of the pre-collected data. We plan to conduct a more detailed analysis on this topic in future work.
>
> &nbsp;
>
> **Q3:** *Scaling issue: if the synthetic dataset is large, the dataset distillation has a high computational cost.*
>
> **A3:** Thank you for bringing this up. While there is a trade-off between performance and efficiency related to the synthetic data size in OBD (see A2 above), OBD is primarily intended to distill a compact synthetic behavioral dataset for efficient policy training. As the synthetic dataset grows larger, the benefits of OBD diminish. Consequently, we focus on optimizing OBD performance within a limited synthetic data budget. We will incorporate this discussion into our revised manuscript.

---

> > ### Comment · Reviewer_VAdy · 2024-08-09
> >
> > I thank the authors for their responses. I would like to remain my score.

---

### Official Review · Reviewer_kzVy · 2024-07-13

**Soundness:** 3
**Presentation:** 3
**Contribution:** 3
**Rating:** 6
**Confidence:** 4

**Summary:**

The paper introduces Offline Behavior Distillation (OBD), a method to synthesize expert behavioral data from sub-optimal reinforcement learning (RL) data to enable rapid policy learning. The authors propose two naive OBD objectives, Data-Based Cloning (DBC) and Policy-Based Cloning (PBC), and introduce a new objective, Action-Value Weighted PBC (Av-PBC). Av-PBC optimizes the weighted decision difference to achieve superior distillation guarantees with linear discount complexity. Theoretical analyses and extensive experiments on multiple D4RL datasets demonstrate that Av-PBC significantly improves OBD performance and convergence speed when compared to the two navie OBD objectives. The paper further shows that the performance of Av-PBC generalize well across architectures of the policy network and optimizers.

**Strengths:**

1. Originality: The paper introduces a novel approach to distill vast sub-optimal RL data into a limited set of expert behavioral data, enhancing policy training efficiency. The formulation of Av-PBC as an OBD objective is innovative and addresses the limitations of the two navie OBD objectives.

2. Quality: The theoretical analysis is robust, providing a solid foundation for the proposed Av-PBC objective. The empirical results are comprehensive, covering multiple datasets and demonstrating significant improvements in OBD performance and efficiency.

3. Clarity:  Overall, the paper is well-organized, with clear explanations of the problem setup, the proposed approach, and results. The theoretical proofs and empirical evaluations are detailed and easy to follow.

4. Significance: The proposed method has broad implications for RL, enabling efficiency pretraining, data privacy protection and prior knowledge construction for downstream tasks.

**Weaknesses:**

1. The dependency on an offline RL algorithm makes the Algorithm 1 fails to claim that OBD could improve training efficiency when compared to direct application of the offline RL with the whole dataset since the training cost of an offline RL is part of the proposed Algorithm 1. Algorithm 1 requires that the an offline RL algorithm should learn pi* and q_{pi*} as an oracle before offline behavior distillation

2. The result has limited discussion on the impact of the quality of the initial dataset on the effectiveness of the Av-PBC objective, limiting its applicability in practice. Adding more insightful discussion and possible guidance for when the Av-PBC could guarantee a performce could promote the adoption of the Av-PBC in practice. For example, an user may ask if it is worth to apply Av-PBC when given a pre-collected dataset.

    * 2.1 Effectiveness Controlled by Dataset Quality: According to Table 1, the quality of the dataset D_off seems to control the effectiveness of the proposed approach. The approach only performs better than BC (whole) on the lowest-quality dataset (M-R). Can the authors discuss this finding in more detail?

    * 2.2 Performance Patterns in Offline Datasets: The explanation for the "interesting phenomenon for Av-PBC" suggests that offline datasets with better state coverage lead to better OBD performance. However, the performance of Medium-Replay and Medium datasets from the HalfCheetah environment contradicts this. Based on their definitions, the M-R dataset has higher state-action coverage than the M dataset, but Av-PBC with the M dataset performs better. It might be worth to discuss these potential OBD performance patterns across offline datasets of different qualities by considering state-action coverage and average trajectory return collectively.

    * 2.3 Av-PBC Ensemble Performance in Hopper Environment: The Av-PBC based ensemble does not improve OBD performance with the M-R and M datasets of the Hopper environment. It makes the paper stronger if the authors provide an explanation to guide potential users in applying the Av-PBC ensemble effectively.

**Questions:**

1. The "Problem Setup" is not straightforward without reading the following sections.

     * 1.1.  Meaning of Capital D in Equation 1: In Equation 1, what does the capital D refer to? After reading the subsection "Problem Setup", it seems it might refer to the set of all trajectories in the environment. Could the authors clarify this?

     * 1.2   Semantics of Surrogate Loss H(pi_theta, D_off):  The surrogate loss is initially introduced here for converting Equation 1 to Equation 2. While the missing clear definition of the surrogate loss makes it difficult to understand that minimizing H(pi_theta, D_off) is equivalent to maximizing the expected return J.  Referring to its definition in the "Problem Setup" would make the paper more readable.

2. Algorithm 1 needs more clarification:

    * 2.1.  Clarification on D_syn in Algorithm 1: In Algorithm 1, should "D_syn" be "B" in the step "Compute H(πθT ,Dsyn)" since the expression on the right side only involves data points from the sampled batch B?

    * 2.2.  Use of GradDescent() in Algorithm 1: Algorithm 1 uses GradDescent() to update the synthesized dataset D_syn. This is understandable for updating policy parameters during behavior cloning, but could the authors clarify the semantics of applying GradDescent() to update D_syn?


3. Examples of Discovered Critical States by Av-PBC: It is mentioned that "distilled behavioral data is also beneficial for explainable RL by showing the critical states and corresponding actions." So, it will make the paper more convincing if some examples of such critical states discovered by Av-PBC could be added to the paper.

**Limitations:**

The authors have addressed some limitations of their work, such as the computational complexity of the bi-level optimization and the performance gap between the synthetic data and the whole offline RL dataset. However, they could further discuss potential limitations of the proposed approach in the following directions:

1.   Generalization to Other Environments: While Av-PBC shows significant improvements on the D4RL benchmark datasets, its performance on more complex and diverse environments like Franka Kitchen and Adroit Manipulation remains unexplored. Future work could investigate the applicability and effectiveness of Av-PBC in these and other challenging environments to understand its generalization capabilities better.

2.   Quality Requirement of Initial Offline Dataset: The paper discusses the performance of Av-PBC under three different qualities of datasets from three Mujoco environments. However, the quality of a dataset can vary widely in practice, potentially leading to failures of Av-PBC in certain scenarios. A discussion on the impact of dataset quality on Av-PBC's performance and practical guidance on applying Av-PBC across different scenarios would enhance its usability and robustness.

---

> ### Author Rebuttal · Authors · 2024-08-05
>
> **Q1:** *The dependency on an offline RL algorithm makes Algorithm 1 fails to claim that OBD could improve training efficiency when compared to direct application of the offline RL with the whole dataset.*
>
> **A1:** Thanks for your comments. Though Algorithm 1 incorporates a complete offline RL process, once the distillation is complete, the distilled data can be reused in various scenarios. **This allows us to train policy networks much more efficiently on distilled data compared to using the large original dataset**. Improving the efficiency of OBD is a key focus for future work, particularly given the challenges of bi-level optimization, as discussed in Limitation. Notably, Algorithm 1 significantly enhances distillation speed compared to naive OBD methods such as DBC and PBC.
>
> &nbsp;
>
> **Q2:** *The result has limited discussion on the impact of the quality of the initial dataset on the effectiveness of the Av-PBC objective, limiting its applicability in practice.*
>
> **A2:** Thanks. Please refer to A2 in the global response.
>
> &nbsp;
>
> **Q2.1:** *The quality of the dataset $D\_\text{off}$ seems to control the effectiveness of the proposed approach in Table 1. The approach only performs better than BC (whole) on the lowest-quality dataset (M-R). Can the authors discuss this finding in more detail?*
>
> **A2.1:** Thanks for your observation. The quality of offline RL data $D\_\text{off}$ directly **affects the expert policy $\pi^\ast$ learned through offline RL algorithm**, thereby impacting the final performance of Av-PBC by the loss in Eq. 7. On the other hand, the policy $\pi$ trained on distilled data typically performs below of $\pi^\ast$ due to optimization errors. **When original data is of lower quality, such as when sampled from sub-optimal or random policy (like M-R), the offline RL algorithm (Offline RL, whole) can learn a much better policy $\pi^\ast$ than BC (whole) because of its specially designed Bellman backup.** This significant performance gap between offRL (whole) and BC (whole) explains why our OBD algorithm performs better than BC (whole) with low-quality $D\_\text{off}$.
>
> &nbsp;
>
> **Q2.2:** *The "interesting phenomenon for Av-PBC" suggests that offline datasets with better state coverage lead to better OBD performance. Therefore, the Medium-Replay (M-R) dataset should have higher state-action coverage than the M dataset, but Av-PBC with the M dataset performs better.*
>
> **A2.2:** Thanks for your comments. We respectfully argue that we only claim that state coverage is crucial for OBD performance, and data quality is also another essential factor, as discussed in A2.1. Although M-R datasets have better state coverage, M datasets have superior data quality, which contributes to its better performance in the Halfcheetah environment. Our additional experiments further explore the impact of state coverage on OBD, revealing that greater state coverage enhances robustness against data noise, which benefits OBD performance. Please refer to A2 in the global response for more details.
>
> &nbsp;
>
> **Q2.3:** *Why does the Av-PBC based ensemble not improve OBD performance with the M-R and M datasets of the Hopper environment?*
>
> **A2.3:** Thanks for pointing this out. The reason the Av-PBC based ensemble does not improve OBD performance with the M-R and M datasets in the Hopper environment is that Hopper is simpler compared to Halfcheetah and Walker2D. Specifically, Hopper has a state dimension of 11 and an action dimension of 3, while Halfcheetah and Walker2D have state dimensions of 17 and action dimensions of 6. In more complex or high-dimensional environments, there is greater policy or model uncertainty, and policy ensembles are more effective at reducing this uncertainty and improving performance. In contrast, the simpler Hopper environment shows fewer improvements with the ensemble approach.
>
> &nbsp;
>
> **Q3:** *What does the capital $D$ in Eq. 1 refer to?*
>
> **A3:** Thanks. The capital $D$ refers to distilled data $D\_\text{syn}$.
>
> &nbsp;
>
> **Q4:** *Missing clear definition of the surrogate loss $H$ makes it difficult to understand that minimizing $H$ is equivalent to maximizing the expected return $J$ in Eq. 1 and Eq. 2.*
>
> **A4:** Thanks for your suggestion. We use the surrogate loss $H$ to approximate the expected return $J$. When $H$ is an appropriate proxy, minimizing $H$ effectively corresponds to maximizing $J$. We will clarify this in the revised paper.
>
> &nbsp;
>
> **Q5:** *In Algorithm 1, "$D_{syn}$" should be "$B$" in "Compute $H(\pi ,D_{syn})$"*
>
> **A5:** Thanks and addressed.
>
> &nbsp;
>
> **Q6:** *In Algorithm 1, could the authors clarify the semantics of applying $\texttt{GradDescent()}$ to update ${D}_\text{syn}$?*
>
> **A6:** Thanks. In the first step of $\texttt{GradDescent}$, we compute the meta-gradient $\nabla_{D_\textbf{syn}}$ via BPTT as shown in Eq. 3. Then we update $D_\text{syn}$ by $D_\text{syn} \rightarrow D_\text{syn} - \alpha \nabla_{D_\text{syn}} $.
>
> &nbsp;
>
> **Q7:** *It will make the paper more convincing if some examples of such critical states discovered by Av-PBC could be added to the paper.*
>
> **A7:** Thanks for your suggestion. Please refer to A1 in the global response.
>
> &nbsp;
>
> **Q8:** *Generalization to Other Environments for proposed Av-PBC*
>
> **A8:** Thanks. We have tested our approach in three widely-used environments of Halfcheetah, Hopper, and Walker2D with various data qualities. Our experiments show that our Av-PBC consistently and remarkably improves the OBD performance compared to other methods in these nine datasets. We will generalize our Av-PBC to more environments in future work to further validate its effectiveness.
>
> &nbsp;
>
> **Q9:** *A discussion on the impact of dataset quality on Av-PBC's performance and practical guidance on applying Av-PBC across different scenarios would enhance its usability and robustness.*
>
> **A9:** Thanks. We provide additional insights on the impact of data quality and state coverage; please refer to A2 in the global response.

---

> > ### Comment · Reviewer_kzVy · 2024-08-11
> >
> > I thank the authors for the detailed and thoughtful response. My concerns have been addressed. And I would like to remain my score.

---

### Official Review · Reviewer_rzRx · 2024-07-14

**Soundness:** 3
**Presentation:** 3
**Contribution:** 3
**Rating:** 5
**Confidence:** 4

**Summary:**

This paper formulates offline behavior distillation in order to enable fast policy learning using limited expert data and thereby leveraging suboptimal RL data. The authors run extensive experiments on D4RL benchmarks to support their findings.

**Strengths:**

1. The linear discount complexity is a vast improvement over the prior quadratic discount complexity.
2. Problem setup and assumptions are written nicely and proof sketches are succinct and concise.
3. The application setting at the end of the paper, sheds light on the scope of the paper.

**Weaknesses:**

There are no major weaknesses, but ablation studies and further corollaries would strengthen the paper.

**Questions:**

I'd like the authors to add their thoughts on how the proposed methods can serve as building blocks towards resolving the limitations mentioned in the paper.

**Limitations:**

There are no significant technical limitations of this paper.

---

> ### Author Rebuttal · Authors · 2024-08-05
>
> **Q1:** *There are no major weaknesses, but ablation studies and further corollaries would strengthen the paper.*
>
> **A1:** thanks for your acknowledgement and suggestion. We have conducted additional empirical studies during the response period to strengthen our paper:
>
> 1. We explored the effect of different sizes of synthetic datasets on OBD performance. The results, presented in Table 2 of the PDF file, indicate that as the size of the synthetic data increases, OBD performance improves. This enhancement is attributed to the larger synthetic datasets conveying more comprehensive information about the RL environment and decision-making processes.
> 2. We examined the role of state coverage in offline RL datasets $\mathcal{D}_\text{off}$ in enhancing OBD. Our empirical findings demonstrate that greater state coverage increases robustness against data noise, which is beneficial for OBD. For a detailed explanation, please refer to A2 in the global response.
> 3. We provided illustrative examples of distilled Halfcheetah behavioral data in Figure 1 of the PDF file. The top row displays the distilled states, while the bottom row shows the subsequent states after executing the corresponding distilled actions in the environment. The figure reveals that (1) the distilled states emphasize "critical states" or "imbalanced states" (for the cheetah) more than "balanced states"; and (2) the subsequent states after taking distilled actions are closer to "balanced states" than the initial distilled states. These examples offer insights into the explainability of reinforcement learning processes.
>
>
> &nbsp;
>
> **Q2:** *I'd like the authors to add their thoughts on how the proposed methods can serve as building blocks towards resolving the limitations mentioned in the paper.*
>
> **A2:** Thanks for your suggestion. In our paper, we highlighted two primary limitations.
> 1. The first limitation involves synthesizing distilled data for reinforcement learning. Our proposed Av-PBC partially addresses this issue by **distilling high-informative behavioral data for imitation learning, one of the offline RL methods**. This approach helps in effectively capturing essential information of the behavioral data, making it a valuable step towards resolving this limitation.
> 2. The second major limitation concerns the high computational resources required for Offline Behavioral Distillation (OBD) due to the bi-level optimization process. Av-PBC offers a significant improvement in this area by enhancing the distillation speed. Specifically, it **reduces the required time to one-quarter** of what is needed by the naive OBD methods like DBC and PBC. This efficiency gain makes Av-PBC a practical and resource-efficient option.
> We will provide a more detailed discussion in the revised version of the paper.

---

> > ### Comment · Reviewer_rzRx · 2024-08-07
> >
> > I thank the authors for their detailed response. My concerns have been clarified. I request to maintain my score.

---

### Author Rebuttal · Authors · 2024-08-05

We appreciate all the reviewers for their enormous efforts and constructive comments. We will make sure to incorporate the parts that you suggested for clarity and reflect your feedback on the revised paper. We have compiled the results of additional experiments related to the reviewers' questions and comments in the attached PDF. The main contents of the file are listed as follows:
- Figure 1: Examples of distilled behavioral data.
- Table 1: Behavioral cloning performance on the offline data **with varying levels of action noise**.
- Table 2: The Av-PBC performance on offline datasets **with different synthetic data sizes**.
- Table 3: Sample size for each offline dataset.

&nbsp;

The following responses present the reviewers' questions and corresponding details of experiments.


**Q1 (by Reviewer kzVy):** *It will make the paper more convincing if some examples of such critical states discovered by Av-PBC could be added to the paper.*

**A1:** Thanks for your suggestion. We have presented some examples of distilled Halfcheetah behavioral data in Figure 1 of the PDF file. The top row displays the distilled states, while the bottom row shows the subsequent states after executing the corresponding distilled actions in the environment. The figure reveals that (1) the distilled states emphasize "critical states" or "imbalanced states" (for the cheetah) more than "balanced states"; and (2) the subsequent states after taking distilled actions are closer to "balanced states" than the initial distilled states. These examples offer insights into the explainability of reinforcement learning processes. We will incorporate this into the revised paper.



&nbsp;

**Q2 (by Reviewer kzVy):**  *The result has limited discussion on the impact of the quality of the initial dataset on the effectiveness of the Av-PBC objective, limiting its applicability in practice.*

**A2:** Thank you for highlighting this point. We have addressed the impact of the quality of original offline RL data  $\\mathcal{D}\_\texttt{off}$ on OBD performance in Sec. 5.1 (Line 277-282), emphasizing the importance of state coverage of $\mathcal{D}_\text{off}$. **The OBD performance is influenced by both state coverage and quality of $\\mathcal{D}\_\texttt{off}$**. Below, we provide additional insights about these two factors:
1.  **Data Quality:** High-quality $\\mathcal{D}\_\texttt{off}$, which includes trajectories with high returns, enhances OBD performance by enabling the learning of an effective expert policy $\pi^\ast$ through offline RL. This in turn supports accurate loss computation of $q_{\pi^*}(s, a)\left(\pi(a \mid s)-\pi^*(a \mid s)\right)^2$ in Eq. 7.

2. **State Coverage:**
    - **Question and Analysis:** An intriguing observation is that the low-quality $\\mathcal{D}\_\texttt{off}$ with better state coverage (such as M-R data) can result in superior OBD performance. The question arises: **how does state coverage benefit OBD**? Our analysis suggests that due to optimization errors in OBD, policies trained on distilled data may not perfectly fit the original dataset $\\mathcal{D}\_\texttt{off}$, but rather with a small error. Thus, **OBD performance, or policy training w.r.t. distilled data, can be likened to behavioral cloning (BC) on a noise-version of $\\mathcal{D}\_\texttt{off}$.**
    - **Experiments and Results:** We conducted BC on the initial data with varying levels of action noise, as detailed in Table 1 of the PDF. The findings show that when there is little to no action noise (noise ratio $\leq 0.05$), higher-quality datasets (M-E) lead to better policy performance compared to datasets with better state coverage (M-R). However, as the noise ratio increases, datasets with better state coverage (M-R) demonstrate greater resilience and outperform the higher-quality datasets (M-E).
    - **Conclusion:** These experiments demonstrate that state coverage enhances robustness against data noise, which partially explains the advantages of state coverage in OBD.

---

### Decision · Program_Chairs · 2024-09-25

**Decision:**

Accept (poster)

**Comment:**

The paper studies the problem of behavior distillation in an offline setting. The reviewers all agree on the contribution of the paper. I recommend acceptance.